# Improved probabilistic regression using diffusion models

Carlo Kneissl[*1,2], Christopher Bülte[*1,2], Philipp Scholl[*1,2], and Gitta Kutyniok[1,2,3,4]

[1]Ludwig-Maximilians-Universität München
[2]Munich Center for Machine Learning (MCML)
[3]University of Tromsø
[4]DLR-German Aerospace Center
 {kneissl, buelte, scholl, kutyniok}@math.lmu.de

## Abstract

Probabilistic regression models the entire predictive distribution of a response variable, offering richer insights than classical point estimates and directly allowing for uncertainty quantification. While diffusion-based generative models have shown remarkable success in generating high-dimensional data, their usage in regression tasks often lacks uncertainty-related evaluation. We propose a novel diffusion-based framework for probabilistic regression where we model the full distribution of the diffusion noise, enabling adaptation to diverse tasks and enhanced uncertainty quantification.

## 1 Introduction

Supervised regression aims at predicting a response variable $\boldsymbol{y} \in \mathcal{Y}$ from covariates $\boldsymbol{c} \in \mathcal{C}$. Classical approaches estimate the conditional mean $\mathbb{E}[\boldsymbol{y} \mid \boldsymbol{c}]$, whereas probabilistic regression models the full predictive distribution $p_{\mathcal{Y}}(\boldsymbol{y} \mid \boldsymbol{c})$ [1, 2], typically via a Gaussian [3, 4]. Recent work has emphasized more flexible, non-parametric alternatives [2, 5].

Diffusion-based generative models have emerged as state-of-the-art approaches for high-dimensional data generation, achieving remarkable results in tasks such as photorealistic image [6] and video synthesis [7].

Recently, diffusion models have been applied to various regression tasks such as depth estimation [8], autoregressive flow prediction [9, 10], and weather forecasting [11, 12], often achieving state-of-the-art performance. Despite their inherently probabilistic nature, evaluations rarely emphasize uncertainty-related metrics or calibration. Recent efforts aim to extract uncertainty estimates from diffusion models [13–15], but typically rely on training multiple networks, incurring substantial computational overhead. Furthermore, the intimate relation between uncertainty quantification and the noise modeling within the diffusion process remains underexplored.

**Contributions:** In this work, we address these limitations by adapting the diffusion process to yield calibrated probabilistic predictions. Building on recent advances from generative modeling [16], we introduce a novel loss for diffusion-based regression models that enables learning a parameterized noise distribution, and offers task-specific trade-offs between expressivity and computational efficiency.

## 2 Background

### 2.1 Probabilistic regression

Let $\boldsymbol{y} \in \mathcal{Y} \subseteq \mathbb{R}^{d_y}$ denote the response variables of interest and $\boldsymbol{c} \in \mathcal{C} \subseteq \mathbb{R}^{d_c}$ the corresponding conditioning variables. Given training data $\mathcal{D} = (\boldsymbol{c}_i, \boldsymbol{y}_i)_{i=1}^{N}$, the goal is to recover the predictive distribution $p_{\mathcal{Y}}(\boldsymbol{y} \mid \boldsymbol{c}, \mathcal{D})$.

### 2.2 Diffusion models

Diffusion probabilistic models (DPMs) aim to learn a target distribution $p_0$ on $\mathbb{R}^d$ from samples by estimating the reverse dynamics of a diffusion process. We follow the non-Markovian formulation of denoising diffusion implicit models (DDIM) [17].

Let $T \in \mathbb{N}$, $\boldsymbol{x}_0 \sim p_0$, and $\beta_{1:T} \in [0,1]^T$ denote a noise schedule. Define $\alpha_t := 1 - \beta_t$ and $\bar{\alpha}_t := \prod_{i=1}^{t} \alpha_i$. The *forward process* produces intermediate states

$$\boldsymbol{X}_t = \sqrt{\bar{\alpha}_t}\boldsymbol{X}_0 + \sqrt{1 - \bar{\alpha}_t}\epsilon_t, \quad \epsilon_t \sim \mathcal{N}(\boldsymbol{0}, \boldsymbol{I}), \quad (1)$$

such that each $x_t$ is a noisy version of $x_0$. For sufficiently small $\bar{\alpha}_T$, $x_T$ is close to a standard normal, providing a tractable prior.

To generate samples, one must approximate the reverse process $p(\boldsymbol{x}_{t-1} \mid \boldsymbol{x}_t)$, which is intractable in general. DPMs approximate it with a latent-variable model

$$p_\theta(\boldsymbol{x}_{0:T}) := p_\theta(\boldsymbol{x}_T) \prod_{t=1}^{T} p_\theta(\boldsymbol{x}_{t-1} \mid \boldsymbol{x}_t), \quad (2)$$

which is a Markov chain that samples from $\boldsymbol{x}_T$ to $\boldsymbol{x}_0$, that is referred to as the *generative process*. For sufficiently small $\beta_t$, the reverse transition $p(\boldsymbol{x}_{t-1} \mid \boldsymbol{x}_t)$ is well approximated by a Gaussian,

---

*Equal contribution

thus allowing to set $p_\theta(\boldsymbol{x}_T) \sim \mathcal{N}(\boldsymbol{0}, \boldsymbol{I})$ and specifying the latent variable model as a neural network, $p_\theta(\boldsymbol{x}_{t-1}|\boldsymbol{x}_t) = \mathcal{N}(\boldsymbol{x}_{t-1}; \boldsymbol{\mu}_\theta(\boldsymbol{x}_t, t), \boldsymbol{\Sigma}_\theta(\boldsymbol{x}_t, t))$.

Training proceeds by minimizing the MSE loss of the noise samples

$$L_{\text{simple}}(\theta) = \mathbb{E}_{t,x_0,\epsilon_t} \left[ \|\epsilon_t - \epsilon_\theta(\boldsymbol{x}_t, t)\|_2^2 \right]. \quad (3)$$

The target distribution is set as $p_0 = p_\mathcal{Y}(\cdot \mid \boldsymbol{c})$, yielding an approximate predictive distribution $p_\theta(\cdot \mid \boldsymbol{c}) \approx p_\mathcal{Y}(\cdot \mid \boldsymbol{c})$.

While diffusion models have shown great success in conditional and unconditional generative modeling, note that due to the objective in Equation (3), the latent variable model only learns to approximate $\epsilon_\theta(\boldsymbol{x}_t, t) \approx \mathbb{E}[\epsilon_t \mid \boldsymbol{x}_t]$ and does not capture information about the full distribution $p(\epsilon_t \mid \boldsymbol{x}_t)$.

## 2.3 Scoring rules

Let $\mathcal{P}$ be a convex set of probability measures on $\mathcal{Y}$ and $\mathbb{P}, \mathbb{Q} \in \mathcal{P}$. A scoring rule $S(\mathbb{P}, \boldsymbol{y})$ [18] is a function that measures the discrepancy between a predictive distribution $\mathbb{P}$ and an observation $\boldsymbol{y} \in \mathcal{Y}$. The expected score is defined as $S(\mathbb{P}, \mathbb{Q}) := \mathbb{E}_{Y \sim \mathbb{Q}}[S(\mathbb{P}, Y)]$. A scoring rule is called *proper* if $S(\mathbb{Q}, \mathbb{Q}) \leq S(\mathbb{P}, \mathbb{Q})$ for all $\mathbb{P}, \mathbb{Q} \in \mathcal{P}$, and *strictly proper* if equality holds if and only if $\mathbb{Q} = \mathbb{P}$. Therefore, strictly proper scoring rules ensure that the true distribution is the unique optimum and have been successfully applied in training neural networks for various tasks [19–21].

# 3 Our Methodology

## 3.1 Learning $p_\theta^\epsilon(\cdot \mid \boldsymbol{x}_t)$

Most diffusion frameworks, following DDPM [22], fix the variance of the noise distribution in each denoising step. This design was originally motivated by two observations: (i) learning the variance often destabilizes training, and (ii) variance modeling showed little benefit for image generation benchmarks, for example, with respect to the FID [23].

However, subsequent work [24] demonstrated that learning the variance improves likelihood estimates, indicating that recovering only the mean is insufficient for faithfully approximating the conditional distribution. Furthermore, the Gaussian approximation of $p(\boldsymbol{x}_{t-1} \mid \boldsymbol{x}_t)$ is only valid when the number of timesteps $T$ is large. Yet, large $T$ is computationally costly, and recent results suggest that using as few as 20–50 steps can yield superior performance in regression tasks [10, 12], particularly with improved noise schedulers and solvers [25, 26].

These observations motivate us to go beyond estimating the first two moments and instead learn the full distribution of $\epsilon_t$. Specifically, we reinterpret

$\epsilon_\theta$: rather than treating it as a point estimate of $\epsilon_t$, we view it as a random variable. This perspective naturally suggests replacing the mean-squared error loss with a criterion that compares probability distributions rather than point predictions. To this end, we adopt the framework of strictly proper scoring rules [20, 27].

## 3.2 Parametrization of $p_\theta^\epsilon(\cdot \mid \boldsymbol{x}_t)$

Concurrent work by Bortoli et al. [16] reached a similar conclusion. They proposed modeling the full posterior distribution $p(\boldsymbol{x}_0 \mid \boldsymbol{x}_t)$ by a neural network and generating samples via noise concatenation to the inputs.

This, however, comes at significant computational cost: multiple samples increase the runtime. Empirically, training is reported to be $1.3\times$ to $7\times$ slower than standard diffusion models [16].

As an alternative to nonparametric sampling-based approaches, we propose to model $p_\theta^\epsilon(\cdot \mid \boldsymbol{x}_t)$ directly through a parametrized distribution. This aligns naturally with scoring rule minimization, since closed-form expressions for the training objective are available, and thereby reducing the training time significantly as we do not need to generate samples. Depending on the choice of parametrization, one obtains a trade-off between computational efficiency and flexibility, which may vary across tasks.

Specifically, we consider for $p_\theta^\epsilon(\epsilon_t \mid \boldsymbol{x}_t)$ the general Gaussian mixture form

$$\sum_{k=1}^K \pi_{\theta,k} \mathcal{N}(\epsilon_t; \boldsymbol{\mu}_{\theta,k}^\epsilon(\boldsymbol{x}_t, t), \boldsymbol{\Sigma}_{\theta,k}^\epsilon(\boldsymbol{x}_t, t)). \quad (4)$$

From this general specification, we highlight three concrete parametrizations that offer a trade-off between the expressivity of the distribution and the simplicity of the model training:

**Univariate Gaussian:** For $K = 1$ and $\boldsymbol{\Sigma}_\theta(\boldsymbol{x}_t, t) = \text{diag}(\sigma_\theta^2(\boldsymbol{x}_t, t))$, $\sigma_\theta^2(\boldsymbol{x}_t, t) \in \mathbb{R}_{>0}^{d_y}$, we obtain a simple baseline.

**Univariate Gaussian mixture.** Setting $\boldsymbol{\Sigma}_{\theta,k}(\boldsymbol{x}_t, t) = \text{diag}(\sigma_{\theta,k}^2(\boldsymbol{x}_t, t))$ with $\sigma_{\theta,k}^2(\boldsymbol{x}_t, t) \in \mathbb{R}_{>0}^{d_y}$ and $K > 1$ we get a Gaussian mixture that approximates arbitrary densities under mild assumptions [1, 28].

**Multivariate Gaussian:** Choosing $K = 1$ with full covariance $\boldsymbol{\Sigma}_\theta(\boldsymbol{x}_t, t)$ allows modeling correlations between the marginals of $\epsilon_t$.

# 4 Conclusion

We propose to learn the full diffusion noise distribution using a Gaussian mixture model, aiming to improve probabilistic regression performance and decrease training time.

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
