# OpenReview forum: "Improved probabilistic regression using diffusion models"
_NLDL.org/2026/Abstracts_Track — NLDL 2026 Abstracts_

### Official Review · Reviewer_D8Ec · 2025-10-29

**Soundness:** 3
**Correctness:** 3
**Rating:** 4
**Confidence:** 3

**Summary:**

In this abstract, authors propose a novel diffusion-based framework for probabilistic regression allowing for uncertainty quantification. In particular, they propose to learn the full diffusion noise distribution using Gaussian Mixture Model with 3 different options: Univariate Gaussian, Univariate Gaussian mixture and Multivariate Gaussian.

**Strengths:**

The motivation seems well supported and the authors appear to be aware of the relevant and recent literature. Their contribution seems relevant to me, as it provides a bridge to the field of uncertainty quantification, thus allowing for more in-depth analysis and estimates.

**Weaknesses:**

The abstract may benefit from revision and further clarification on several points:

- W1: Section 2.3 takes up quite a lot of space and could be summarized given the accessible content.
- W2: In lines 133–134 it is not clarified which scoring rule function is used, although it is underlined as a contribution.
- W3: It is not clear the connection of $p_{\theta}^{\epsilon} (\epsilon_t,x_t)$ with paragraph 2.2
- W4: It is not discussed the trade-offs of the different parametrizations (neither discussed nor minimally justified).

I believe that the message and contribution could be improved to ensure dialogue with researchers from other fields, which is the purpose of the conference.

---

### Official Review · Reviewer_pGs5 · 2025-10-30

**Soundness:** 3
**Correctness:** 3
**Rating:** 4
**Confidence:** 4

**Summary:**

This work explores a promising direction for improving the calibration of probabilistic predictions within diffusion models. While these models have demonstrated strong generative performance, as the authors mention, there has been comparatively little focus on quantifying their uncertainty. I see their approach aims to enhance reliability by better aligning predicted probabilities with true outcome frequencies, contributing to more trustworthy and interpretable diffusion-based systems.

**Strengths:**

They manage to cover meaningful mathematical content in the short format, which makes it easier to see how it can move beyond just a conceptual idea.

**Weaknesses:**

This is an abstract, so I do not see major weaknesses; perhaps more literature on existing work could be included, but it would compromise the methods.

---

### Official Review · Reviewer_ka24 · 2025-11-03

**Soundness:** 4
**Correctness:** 3
**Rating:** 4
**Confidence:** 2

**Summary:**

This abstract outlines a novel approach to extending probabilistic regression within the domain of diffusion models, through the learning of a parameterised distribution $p_{\theta}^{\epsilon}(. | x_{t})$. The authors outline a plan to use scoring rule minimisation, through proposed parametrisation methods: Univariate Gaussian, Univariate Gaussian mixture and multivariate Gaussian, depending on the task chosen. The main contribution revolves around achieving this mean and variance modelling through an explicit noise distribution that would allow for likelihood estimates to be extracted while remaining computationally efficient.

**Strengths:**

1. The authors' overview of the relevant background supports their proposed method. Where the background section successfully highlights the theory behind both probabilistic regression and diffusion models. Furthermore, the work successfully provides the backing knowledge as to why traditional diffusion methods fix variance and only recover the mean, and why recovering the full conditional distribution requires this change.
2. A possible baseline and relevant work are presented through the work done by Bortoli et al.
3. All formulas are clearly outlined, with terms defined adequately to understand the core propositions of the work.
4. Closed-form data generation clarifies how the authors plan to generate the ground distribution data.
5. The work compellingly presents an approach using proper scoring rules to model both mean and variance through a parametrised distribution, avoiding the computational cost of modelling the distribution by Neural Network concatenation techniques.
6. Computational improvements are clear in comparing the large time-steps $T$ required for traditional diffusion models to the gains that could be made with lower time-steps.

**Weaknesses:**

1) While the closed-form generation of the data is mentioned, the exact nature of the data and application is not touched on. Inclusion of such examples would clarify the various use cases of the Gaussian parameterisations put forward.
2) Line 43 indicates a novel loss being proposed, while line 134 (and line 149) clarifies that the framework of strictly proper scoring rules would be used. It's unclear if a novel loss that is not proposed here will be used or whether the contribution is with the combination of the parameterised distribution.
3) It is a bit unclear how the method proposed will be computationally cheaper than what is proposed by Bortoli et al. Furthermore, while the computational improvements from reducing the number of timesteps $T$ for Gaussian approximation are yet to be seen, it also leads me to wonder how much more efficient learning a parametrised distribution would be, given that more parameters may be needed to train such a model.

---

### Decision · Program_Chairs · 2025-11-05

**Decision:**

Accept

**Comment:**

The abstract is of interest to the community and should be presented at the conference.